# Rapid Identification and Analysis of Ochratoxin-A in Food and Agricultural Soil Samples Using a Novel Semi-Automated In-Syringe Based Fast Mycotoxin Extraction (FaMEx) Technique Coupled with UHPLC-MS/MS

**DOI:** 10.3390/molecules28031442

**Published:** 2023-02-02

**Authors:** Karthikeyan Prakasham, Swapnil Gurrani, Jen-Taie Shiea, Ming-Tsang Wu, Chia-Fang Wu, Yi-Jia Ku, Tseng-Yu Tsai, Hung-Ta Hua, Yu-Jia Lin, Po-Chin Huang, Gangadhar Andaluri, Vinoth Kumar Ponnusamy

**Affiliations:** 1PhD Program in Environmental and Occupational Medicine & Research Center for Precision Environmental Medicine, Kaohsiung Medical University (KMU), Kaohsiung City 807, Taiwan; 2Department of Medicinal and Applied Chemistry, Kaohsiung Medical University (KMU), Kaohsiung City 807, Taiwan; 3Department of Chemistry, National Sun Yat-Sen University, Kaohsiung City 804, Taiwan; 4Department of Family Medicine, Kaohsiung Medical University Hospital, Kaohsiung Medical University (KMU), Kaohsiung City 807, Taiwan; 5International Master Program of Translational Medicine, National United University, Miaoli 36063, Taiwan; 6Research and Development Center, Great Engineering Technology (GETECH) Corporation, No. 392, Yucheng Rd., Zuoying District., Kaohsiung City 813, Taiwan; 7National Institute of Environmental Health Sciences, National Health Research Institutes (NHRI), Miaoli 35053, Taiwan; 8Civil and Environmental Engineering Department, College of Engineering, Temple University, Philadelphia, PA 19122, USA; 9Department of Medical Research, Kaohsiung Medical University Hospital (KMUH), Kaohsiung City 807, Taiwan

**Keywords:** coffee, tea, ochratoxin-A, micro-QUECHERS, in-syringe, UHPLC-MS/MS

## Abstract

In this work, a fast mycotoxin extraction (FaMEx) technique was developed for the rapid identification and quantification of carcinogenic ochratoxin-A (OTA) in food (coffee and tea) and agricultural soil samples using ultra-high-performance liquid chromatography-tandem mass spectrometry (UHPLC-MS/MS) detection. The FaMEx technique advancement is based on two plastic syringes integrated setup for rapid extraction and its subsequent controlled clean-up process. In the extraction process, a 0.25-g sample and extraction solvent were added to the first syringe barrel for the vortex-based extraction. Then, the extraction syringe was connected to a clean-up syringe (pre-packed with C18, activated carbon, and MgSO_4_) with a syringe filter. Afterward, the whole set-up was placed in an automated programmable mechanical set-up for controlled elution. To enhance FaMEx technology performance, the various influencing sample pretreatment parameters were optimized. Furthermore, the developed FaMEx method indicated excellent linearity (0.9998 and 0.9996 for coffee/tea and soil) with highly sensitive detection (0.30 and 0.29 ng/mL for coffee/tea and soil) and quantification limits (1.0 and 0.96 for coffee/tea and soil), which is lower than the toxicity limit compliant with the European Union regulation for OTA (5 ng/g). The method showed acceptable relative recovery (84.48 to 100.59%) with <7.34% of relative standard deviation for evaluated real samples, and the matrix effects were calculated as <−13.77% for coffee/tea and −9.7 for soil samples. The obtained results revealed that the developed semi-automated FaMEx/UHPLC-MS/MS technique is easy, fast, low-cost, sensitive, and precise for mycotoxin detection in food and environmental samples.

## 1. Introduction

Coffee and tea are common non-alcoholic beverages that are extensively consumable drinks globally. Coffee and tea comprise several families of chemical compounds to promote well-being [1]. However, mycotoxins can also be found in coffee and tea samples. Mycotoxins are produced by various fungal species, such as Aspergillus, Penicillium, and Fusarium species. Pre- and post-harvest contamination with these fungal species, particularly Aspergillus and Penicillium, can grow under low water activity storage conditions [2]. Mycotoxins can cause acute and chronic toxicity to human and animal health by contaminating various food, mainly coffee and tea. Different coffee and tea samples have been detected with aflatoxins, fumonisins, and ochratoxin A (OTA). Among them, OTA is a potentially threatening toxicant produced by various fungal species. The chemical structure of the OTA is illustrated in Figure 1, representing the most common mycotoxin found in coffee and tea. The fungus Aspergillus ochraceus and Penicillium verrucosum are the primary producers of OTA, and it shows relevant toxicity due to their nephrotoxicity and hepatotoxicity nature to humans and animals [3,4,5]. Moreover, fungal growth in the decayed plant or disposal of unused or expired tea and coffee products in landfills often contaminates the nearby landfill soils [6]. The IARC (International Agency for cancer research) classifies OTA as possibly carcinogenic to humans; European Union sets the toxicity limit as 5 ng/g in coffee products [7].

Due to the severe toxicity of OTA in coffee, tea, and soil samples, it is essential to develop a rapid and sensitive analytical method to quantify OTA for routine food safety testing. Remarkably, the coffee and tea matrix is highly complex due to the presence of Maillard reaction products, polysaccharides, proteins, fats, caffeine, and polyphenols, including flavonoids and phenolic acids [8,9,10,11,12]. These chemicals present in the coffee and tea matrix are readily soluble in the extraction solvent and interfere with target analytes during the instrumental analysis, thereby affecting the analysis and results accuracy. Many research studies developed various extraction steps, such as solid-phase extraction (SPE) cartridges, immuno-affinity SPE columns, floating organic solvents, and dispersive and non-dispersive solvent-based liquid-liquid extraction, to perform in coffee and tea products. However, these methods are time-consuming, laborious, require lots of toxic solvents for extraction, expensive clean-up sorbents, and need highly skilled analysts [13].

Recently, the sample pretreatment method, Quick, Easy, Cheap, Effective, Rugged, and Safe (QuEChERS), has more advantages for the determination of residual toxins in food, environmental and biological samples [14]. It is significantly advantageous to analyze various toxins with different polarities by their unique clean-up sorbents. The clean-up process was designed for QuEChERS extraction based on the adsorption of sample matrixes instead of target analytes. A primary-secondary amine and C18 were mainly used as the SPE clean-up sorbent to remove sterols, pigments, non-polar, and acidic compounds. In addition, anhydrous magnesium sulfate was used to remove residual moisture content in the sample [15]. Furthermore, the technique continues to gain popularity through various modifications by developing appropriate methodological kits for either QuEChERS extraction or dispersive-SPE clean-up. However, these methods consist of several drawbacks, such as large solvent volume, long extraction time by vigorous shaking, long centrifugation, transferring solvents to dSPE cleanup, expensive sorbents, and less sensitivity.

In this study, we report a facile, fast mycotoxin extraction (FaMEx) procedure using two integrated syringes setup for extraction and clean-up to analyze mycotoxin (OTA) in coffee, tea, and agricultural soil samples using LC-MS/MS. The factors enriching the extraction efficiency of OTA in these samples (higher complexation with the co-matrixes) by the proposed method were completely examined and optimized. The fully validated analytical methodology was applied for the real-time coffee, tea, and soil sample analysis.

## 2. Results and Discussion

The fast mycotoxin extraction and clean-up performances for OTA analysis were examined and optimized, particularly various influencing parameters, including extraction solvent, extraction solvent volume, extraction time, SPE clean-up sorbent type, sorbent amount, and clean-up flow rate. All the parameters were optimized by the one variable at a time approach under triplicate analysis, and the results were discussed as follows.

### 2.1. Optimization of the Extraction Process

#### 2.1.1. Effect of Extraction Solvent

The extraction solvent is a significant factor that affects the extraction performance in FaMEx. Therefore, selecting a suitable extraction solvent is essential to avoid a negative influence on the extraction process. A proper solvent must possess high extraction capacity and good chromatographic behavior for OTA separation and detection [16]. Five organic solvents were screened, including acetonitrile (MeCN), ethyl acetate (EA), acetone (ACTN), methanol (MeOH), and isopropyl alcohol (IPA), and were tested for OTA extraction and analysis under the developed method.

2.0 mL of extraction solvent was taken and vortexed for 2.5 min at 3000 rpm, and the extraction solvent was passed through the FaMEx-SPE cartridge containing 50 mg C18, 75 mg AC, and 1000 mg anhydrous MgSO_4_ for the clean-up process. Acetonitrile (MeCN) shows maximum extraction recovery due to the least polar nature compared to the other selected solvents and which shows >90% with no matrix peaks interferences, as illustrated in Figure 2a. EA solvent extraction results indicated poor chromatographic behavior and improper peak shape (non-symmetrical) for OTA. MeOH, IPA, and ACTN extractions resulted in intense brown color after extraction (As shown in Appendix A) due to the higher affinity towards the tannin, Millard products, chlorophylls, and flavonoids. This is because of the polar nature of MeOH and IPA, and these solvents result in the lowest extraction recovery for OTA due to high matrix influences. Thus, MeCN was eventually chosen as an extractant for subsequent analysis.

#### 2.1.2. pH of the Extraction Medium

Ochratoxin-A (a weak acid) compound with two pKa values of 4.4 and 7.3 due to the presence of phenylalanine, and isocoumarin functional group [17,18,19]. Because of the typical nature of OTA, the pH condition of the extraction medium needs to be optimized to achieve the maximum extraction from the sample matrix. Hence, the addition of 0.5 mL of various concentrations of acid-base solution into the sample matrix; such as 1% formic acid (pH ≤ 1), 0.5% formic acid (pH = 1), 0.1% formic acid (pH = 2.9), 1% ammonia (pH ≥ 14), and 0.1% ammonia (pH = 9) solution were tested. Experimental results show that the addition of different concentrations of ammonia solutions to the sample matrix leads to poor extraction efficiency. As expected, based on the pKa values of OTA, adding 0.5 mL of 0.1% formic acid indicates maximum extraction efficiency over 1% formic acid due to the maximum conversion of analyte to neutral forms at acidic pH. As a result, the addition of 0.5 mL of 0.1% formic acid into the sample matrix has been selected for subsequent analysis as shown in Appendix A.

#### 2.1.3. Effect of Extraction Solvent Volume

Various extraction solvent (MeCN) volumes (0.5 to 3 mL) were tested to obtain the highest extraction capacity for OTA under the proposed method. The MeCN volumes from 0.5 mL to 1 mL indicated inadequate solvent for complete dispersion of 0.25 mg of the coffee/tea/soil sample matrices and showed low extraction capacities of OTA. 2 mL of MeCN showed maximum extraction capacity for OTA under the developed method. Volumes greater than 2 mL of MeCN show a decreased peak intensity due to the dilution of the target analyte (OTA) in the extraction solvent medium. Hence, 2 mL of extraction solvent (MeCN) volume has been fixed for the further optimization process.

#### 2.1.4. Effect of Extraction (Vortex) Time

The extraction time is an essential factor in achieving the highest extraction of OTA from the sample matrix. Inversely, coffee, tea, and soil products are more affluent in Millard products, chlorophylls, lipids, proteins, flavonoids, caffeine, phenolic acids, and numerous degradation products; hence increase in the extraction time leads to co-extract the matrix components in the sample matrix tends to poor performances. Various extraction times (from 0.5 to 3 min) under a vortex agitation mixer were examined to achieve higher OTA extraction with low matrix interferences. Figure 2b indicates that extraction recovery of OTA was increased from 1 to 3 min and remained constant at 2.5 min. Therefore, 2.5 min of extraction (vortex agitation) time is ideal for extracting maximum OTA from the sample matrix, and it can able to avoid the less acetonitrile soluble matrices such as lipids and unsaturated fatty acids [20]. Hence, 2.5 min of extraction time was selected for the FaMEx-SPE clean-up optimization process.

### 2.2. Optimization of the Clean-Up Process

#### 2.2.1. Effect of Sorbent and Sorbent Amount

The SPE clean-up process is crucial to remove co-extracted interferences from extraction solvent because of highly colored Millard products, lipids, proteins, flavonoids, and phenolic acids in the coffee and tea samples. Similarly, soil products contain numerously non-degradable and degradable chemicals from various environmental and anthropogenic sources. Four absorbents (AC, AG, GCB, and C18) were selected for SPE clean-up to remove the co-extracted interferences from the selected extraction solvent. Results showed that AC shows maximum elimination of co-extracted interferences/co-matrix clean-up from the extractant and indicated the highest extraction recovery for the target OTA. In addition, AC offers good adsorption/affinity toward the co-eluents from the coffee, tea, and soil sample matrices. Other sorbents (AG, GCB, and C18) failed to remove the colored extracts from the coffee, tea, and soil samples. However, adding C18 shows the efficient removal of the sample matrix as reported [21,22,23]. Therefore, AC and C18 were selected as clean-up sorbents to improve the proposed extraction performances of the FaMEx technique. Different amounts (10–100 mg) of AC and C18 were tested by one variant at a time. 75 mg of AC shows the highest extraction recovery for OTA because of the maximum elimination of inferences such as high molecular weight Millard products and colored components in the extraction solvent. On the other hand, 50 mg of C18 showed maximum elimination of fats and indicated a maximum extraction recovery for OTA, as shown in Figure 3a,b. Therefore, 75 mg of AC and 50 mg of C 18 were chosen as the optimum sorbents for the SPE clean-up process.

#### 2.2.2. Effect of Water Content

In the clean-up process, the elimination of water content from the extraction solvent plays a crucial role in increasing the extraction recovery of OTA. Various amounts (200 to 1200 mg) of anhydrous MgSO_4_ along with SPE clean-up sorbents (50 mg C18 and 75 mg AC) were assessed to achieve the maximum extraction recovery for OTA in the proposed method. Figure 3c shows that maximum extraction recoveries were obtained for OTA using ≥1000 mg of anhydrous MgSO_4_. Therefore, 1000 mg of anhydrous MgSO_4_ was chosen as the optimum amount for the FaMEx procedure.

#### 2.2.3. Optimization of Plunger Speed

In the FaMEx-SPE clean-up procedure, the time of interaction between the sorbent and analyte is a critical factor in achieving maximum clean-up and getting higher extraction recovery for OTA [16,17,24]. Therefore, to achieve maximum clean-up efficiency, the optimized quantity of sorbents was homogeneously mixed and packed between the two frits in a 6 mL syringe (75 mg AC, 50 mg C18, and 1000 mg anhydrous MgSO_4_), and the flow rates were examined under various plunger speed from 0.5 to 3.0 mL/min to study the maximum clean-up performance under the developed automated SPE clean-up process. Increasing the plunger speed above 1.0 mL/min showed a decreasing trend in the extraction efficiency of OTA. Therefore, both 0.5 and 1.0 mL/min offers the maximum extraction efficiency for OTA under the proposed method. However, a 0.5 mL/min flow rate was selected (Figure 3d) because of better color removal and maximum extraction efficiency compared to 1.0 mL/min.

### 2.3. Method Performances

#### 2.3.1. Analytical Performances of the Developed Method

Optimal conditions for extraction and clean-up were well studied and optimized one variant at a time. The method was validated to quantitate OTA using LC-MS/MS (optimized conditions: 2 mL solvent, 2.5 min vortex time, SPE Package: 1000 mg MgSO_4_, 75 mg AC, and 50 mg C18 and 0.5 mL/min plunger speed). The calibration was performed using different spiking levels in the sample matrices (coffee/tea/soil). The matrix-matched calibration equation plots resulted in excellent linearity for the target OTA between the 1–100 ng/g calibration range with correlation coefficients of 0.9998 and 0.9996 for coffee/tea and soil samples, respectively. In addition, the LOD and LOQ were calculated for the method calibration by applying the signal-to-noise ratio of the method calibration range and observed LOD for coffee/tea and soil were 0.30 and 0.29 ng/g and LOQ for coffee/tea and soil were 1.0 and 0.96 ng/g, respectively.

#### 2.3.2. Matrix Effect

Matrix effects (MEs) occur due to the residual of the sample matrix or interference components present in the extractant after the clean-up process. The co-elution of the sample matrix components in the same retention with the target analyte makes the signal suppression or enhancement (SSE) in the ESI source. Therefore, the evaluation of the ME, signal area of the three levels of calibration point obtained from MeCN, and the same three levels of the matrix-matched solvent signal area were compared and calculated using Equation (1).
(1)$$\mathrm{SSE}\;\left(\%\right)=100\times (\frac{Signal\;area\;of\;matrix\;matched\;solvent}{Signal\;area\;of\;blank\;MeCN})$$

Experimental results show that the matrix-matched solvent signal areas were less than the blank solvent for the coffee and soil matrix. Therefore, it indicates that signal suppression is due to the co-elution of the sample matrix, and the average calculated matrix effect percent was found to be −13.77 ± 7.31 for coffee and −9.7 ± 6.23 for soil samples.

#### 2.3.3. Real Sample Analysis

The fully validated method was applied to quantitate OTA in three coffee, tea, and soil samples. Coffee and tea samples were purchased from a convenience store, and soil samples were collected in the agricultural area of Kaohsiung, Taiwan. Test results were presented in Table 1, indicating that OTA was not present or below the detection limit of coffee, tea, and soil samples. These results confirm the zero exposure of the OTA-producing fungi in coffee/tea/soil samples collected from Kaohsiung, Taiwan. Moreover, an analysis of three spiking levels in sample matrices was carried out to examine the accuracy of the developed method. The obtained results are listed in Table 1 in terms of recovery (calculated using Equation (2), and the recoveries ranged from 84.48% to 100.59% for both coffee, tea, and soil samples, with an RSD of less than 7.34% for triplicate analysis. These results exhibit that the excellent extraction recoveries over all samples are acceptable. The total ion chromatograms of blank, 10 ng/g OTA extracted from coffee, tea, and soil samples are shown in Figure 4. These results indicate that the developed extraction cum SPE clean-up method coupled with LC/MS/MS method is fast, more reliable, highly sensitive, and semi-automated than previously reported methods for quantitation of OTA coffee samples.
(2)$$\mathrm{Recovery}\;\left(\%\right)=\frac{Conc.\;in\;extraction\;solvent\times Volume\;of\;extraction\;solvent\;}{Sample\;amount\times Conc.\;spiked\;in\;sample\;}\times 100$$

### 2.4. Comparison with Previously Reported Methods

To further demonstrate the advantages of the established method coupled with the UHPLC-MS/MS method to quantitate OTA from coffee/tea/soil samples, the critical parameters of the extraction procedures, such as solvent volume, extraction time, and LODs, utilized in the established method were compared with other previously reported methods for quantitating OTA. All the results are compared in Table 2. The developed method requires a low-organic solvent (2.0 mL), short extraction time, minimal steps, and good recovery when compared to previously reported results (Table 2). Thus, the developed method is simple, rapid, efficient, and more cost-effective than the previously reported methods.

## 3. Materials and Methods

### 3.1. Chemicals and Reagents

HPLC-grade solvents such as acetonitrile (MeCN), isopropyl alcohol (IPA), and methanol (MeOH) were purchased from Aencore Chemicals Pvt. Ltd. (Surrey Hills, Australia). Ethyl acetate (EA) and acetone (ACTN) were purchased from Echo Chemicals (Maioli City, Taiwan). Formic acid was obtained from Fisher Chemical (Leicestershire, UK). Ammonium sulfate ((NH_4_)_2_SO_4_), sodium sulfate (Na_2_SO_4_), sodium chloride (NaCl), sodium hydroxide (NaOH), and magnesium sulfate (MgSO4) were purchased from Xilong Scientific Co., Ltd. (Shanghai, China). Graphitized carbon black (GCB) and octadecylsilane (C18) were purchased from Phenomenex (Torrance, CA, USA). Activated carbon (AC) and amorphous graphite (A.G.) were purchased from Fischer. Luer lock syringes were purchased from Terumo Taiwan Medical Co., Ltd., Taipei City, Taiwan.

### 3.2. Instrument Conditions

#### 3.2.1. UHPLC Conditions

Analysis of OTA was performed using a UHPLC Nexera-i 2040C 3D system (Shimadzu, Tokyo, Japan). The C18 column (250 × 4.6 mm, 3 μm; ACE column) stationary phase was used to separate OTA. 0.1% Formic acid in water (mobile phase A) and acetonitrile (mobile phase B) were used as mobile phases. The isocratic condition was used to achieve a better separation with mobile phase A & B composition (20:80, *v*/*v*) with a flow rate of 0.6 mL. The sample injection volume was 5 μL, and the column oven was kept constant at 40 °C.

#### 3.2.2. Mass Spectrometer Conditions

A Shimadzu LCMS-8045 triple quadrupole mass spectrometer (Shimadzu, Tokyo, Japan) equipped with an ESI source under positive ion mode was applied to analyze OTA using MRM mode. The heat block temperature at the ion source was set at 300 °C, the DL temperature at 350 °C, the interface temperature at 300 °C, the nebulizing gas flow rate at 3 L min^−1^, and the heating gas flow rate at 10 L min^−1^ for the MS/MS analysis. The Collision cell energy was optimized under MRM auto-optimization mode using OTA standard solution to identify maximum MS/MS signals at voltage −22.0 and −38.0 for *m*/*z* 404.2 to *m*/*z* 239.0 and *m*/*z* 404.2 to *m*/*z* 221.0.

#### 3.2.3. Automated Plunger Device Set-Up for the Clean-Up Process

The modified commercial high-pressure plunger pump machine was adapted for SPE clean-up application using a programming module to alter the plunger speed from 0.5–5 mL/min for analytical performance, as illustrated in Figure 5. The two high-pressure plungers at positions (7 and 9) were mounted as defined in the instrumental configuration in Figure 3. Both plungers were connected to the panels’ positions at 6 and 8 and were linked to the top-down mobile frame at positions (1, 2, and 3) for extraction tubes and solvent collection tubes, respectively. Moreover, the set-up is placed in a vertical stand position at 5. The entire device was operated and controlled through the digital LCD display position at 4. The other parts of the experimental set-up were the syringe column (11), plastic plungers (10), SPE disk-shaped frits (12), stopper (13), sorbent packed syringe with filter connecter (14), 0.22 µm PFTE filter (15), and the connection of two syringes, as graphically illustrated in Figure 5.

### 3.3. Semi-Automated FaMEx Procedure

The overall extraction procedure of OTA is graphically illustrated in Figure 6. In detail, 250 mg homogenized sample (coffee/tea or soil) and 0.5 mL of water, and 2 mL of MeCN were added into a 10 mL syringe with a bottom stopper. Then, the syringe setup was vortexed for 2.5 min at 3000 rpm speed. Then, the syringe set-up was connected to the FaMEx-SPE cartridge packed with a homogeneous mixture of MgSO_4_, AC, and C18, in between the two ceramic frits. Then, the integrated dual-syringe step was placed on the automated plunger device, and the piston was programmed to press the extracted solution through the sorbents at the piston’s speed (10 Hz, i.e., 0.5 mL/min). Finally, the obtained clean extract was taken for UHPLC-MS/MS analysis.

## 4. Conclusions

The newly developed FaMEx method was applied to analyze mycotoxin (OTA) in coffee, tea, and soil samples. The presented extraction method is simple, features low-solvent consumption, and a few-step procedure makes this protocol more efficient for real-time applications. In addition, the controlled, semi-automated SPE clean-up process using the newly modified auto-plunger increases the accuracy of the method. Moreover, the proposed FaMEx method shows less matrix effect and acceptable extraction recovery with good precision values for complex food and environmental samples. Therefore, the present method can be applied as a potential alternative methodology in standard laboratories for the analysis of mycotoxins in food and environmental samples.

## Figures and Tables

**Figure 1 molecules-28-01442-f001:**
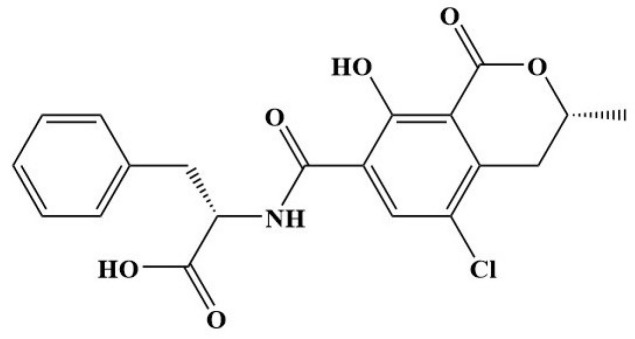
Chemical structure of target analyte (Ochratoxin-A, OTA) of this study.

**Figure 2 molecules-28-01442-f002:**
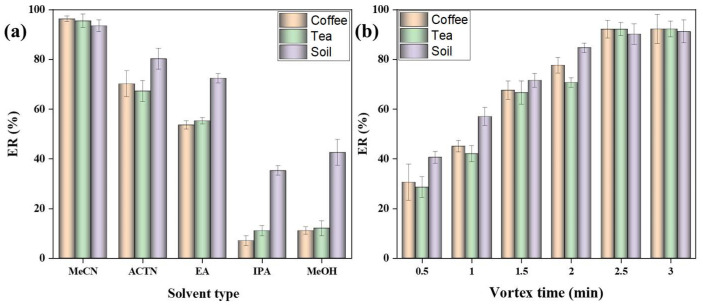
(**a**) Selection of extraction solvent, and (**b**) Selection of extraction time for the extraction of OTA from coffee samples. *Extraction conditions:* 250 mg sample, 2.0 mL solvent, 2.5 min of vortex extraction, and clean-up under the flow rate of 0.5 mL/min using the 1000 mg MgSO_4_, 75 mg AC and 50 mg C18 with three replicate analyses.

**Figure 3 molecules-28-01442-f003:**
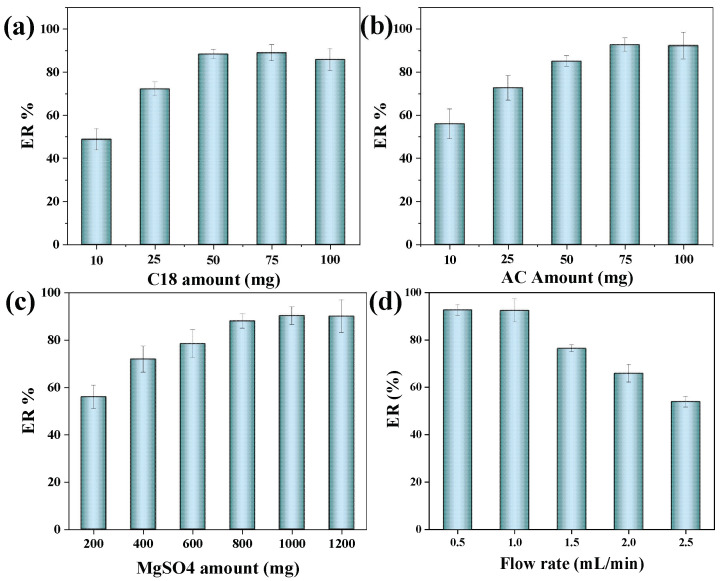
(**a**) Selection of C 18 amount, (**b**) Selection of AC amount and (**c**) Selection of MgSO_4_ amount, and (**d**) Selection of plunger speed for the extraction of OTA from coffee samples. Extraction conditions: 250 mg coffee, 0.5 mL water, 2.0 mL solvent, 2.5 min of vortex extraction, and clean-up under the flow rate of 0.5 mL/min using the 1000 mg MgSO_4_, 75 mg AC and 50 mg C18 with three replicate analyses.

**Figure 4 molecules-28-01442-f004:**
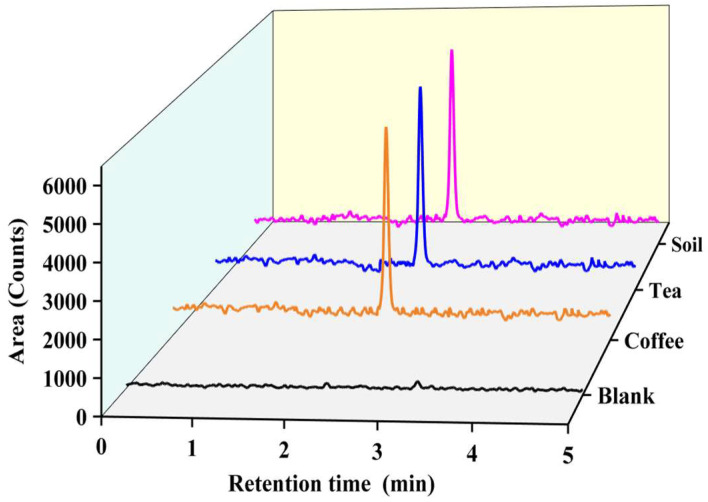
Total ion chromatogram of blank solvent, and 1 ng/g OTA extracted from coffee, tea, and soil matrix.

**Figure 5 molecules-28-01442-f005:**
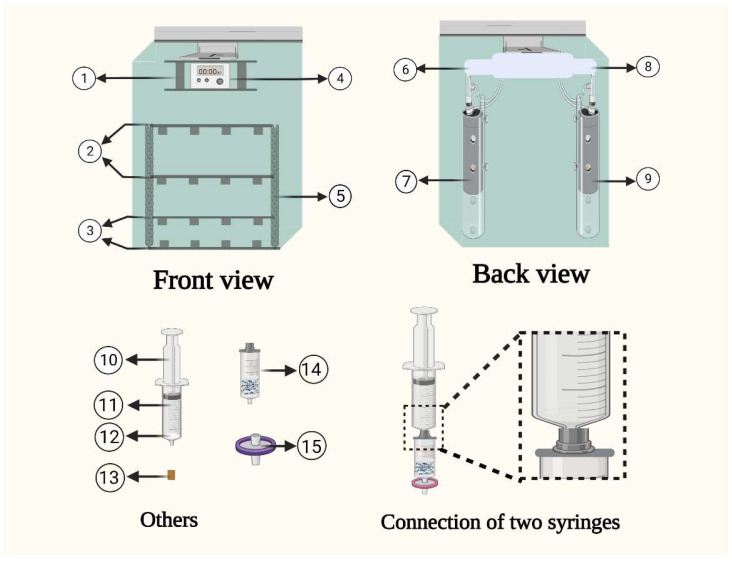
Illustration of the design of the developed semi-automated instrument for the FaMEx clean-up process. Parts numbers in this figure: 1. Plunger rods, 2. Extraction syringe holder, 3. Clean-up syringe holder, 4. Monitor, 5. Syringe stand, 6 and 8. Plunger controller, 7 and 9. Motor, 10. Plunger, 11. Extraction syringe barrel, 12. Hydrophobic frits, 13. Stopper for extraction syringe, 14. Clean-up syringe, and 15. Filter.

**Figure 6 molecules-28-01442-f006:**
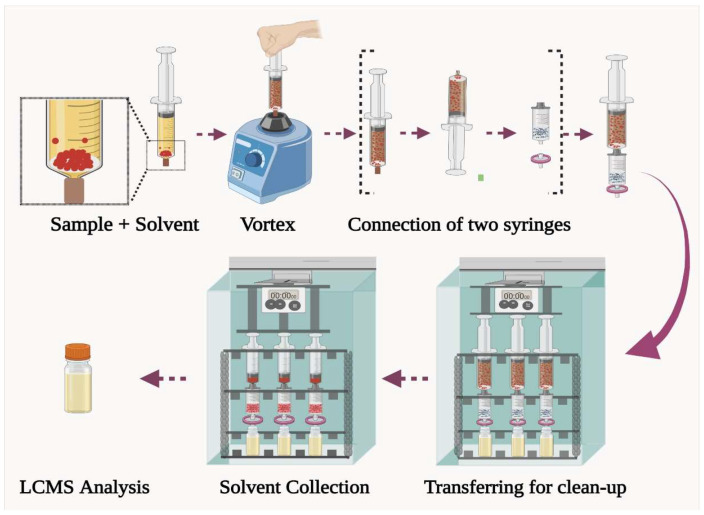
Graphical representation of FaMEx extraction procedure.

**Table 1 molecules-28-01442-t001:** Real sample analysis (coffee, tea, and soil) by the proposed method (n = 3).

Sample Type	Spiked Conc (ng/g)	Intraday Recovery (%)	RSD %	Interday Recovery (%)	RSD %
Coffee sample-1	0	BQL	-	BQL	-
	1	82.48	3.06	88.05	6.31
	10	85.55	1.15	85.84	2.75
	25	90.37	7.34	94.95	6.81
Coffee sample-2	0	BQL	-	BQL	-
	1	92.14	5.06	90.29	4.46
	10	87.48	2.74	84.14	1.86
	25	82.82	3.34	91.56	5.26
Coffee sample-3	0	BQL	-	BQL	-
	1	83.84	1.67	85.87	6.82
	10	82.50	1.11	98.85	2.89
	25	95.32	4.77	91.45	2.35
Tea sample-1	0	BQL	-	BQL	-
	1	86.02	5.75	88.64	2.06
	10	100.59	4.18	85.24	4.91
	25	90.24	2.48	86	2.52
Tea sample-2	0	BQL	-	BQL	-
	1	85.86	3.60	89.22	3.39
	10	87.58	1.07	92.88	5.46
	25	86.29	1.59	90.60	6.96
Tea sample-3	0	BQL	-	BQL	-
	1	84.73	3.58	98.57	2.81
	10	91.54	1.32	86.59	6.63
	25	88.27	1.22	86.54	2.71
Soil sample-1	0	BQL	-	BQL	-
	1	86.92	3.41	90.42	5.36
	10	90.76	5.26	85.06	3.67
	25	92.42	4.78	97.81	6.42
Soil sample-2	0	BQL	-	BQL	-
	1	89.53	3.80	95.89	3.45
	10	93.68	4.96	87.66	6.41
	25	87.27	5.88	88.44	2.90
Soil sample-3	0	BQL	-	BQL	-
	1	89.46	2.74	92.48	3.69
	10	87.67	6.45	97.52	5.63
	25	97.51	5.29	104.61	6.05

**Table 2 molecules-28-01442-t002:** Comparison of the developed method with previously reported methods.

Analyte Studied	Extraction Method	Analytical Technique	Solvent Volume (mL)	Extraction Time (min)	Sample Amount	LOD (ng/g)	Ref.
OTA	SPE	LC-FD	100	>60	20 ^#^	0.4	[25]
OTA	MWE	LC-FD	50	20	2.5 ^#^	5	[26]
OTA, AFs, DON, ZEL, FB1, FB2, T-2 & HT-2	QuEChERS	LC-MSMS	10	20	5 ^#^	-	[27]
AFs & OTA	QuEChERS-SPE	LC-MSMS	40	35	5 ^#^	0.5	[28]
OTA	DLLME–SFO	LC-MSMS	-	20	5 *	0.5	[5]
AFs, DONs, NIV, T-2, HT-2, ZEA, OTA & ENNs	DLLME	LC-MSMS	4.14	25	5 ^#^	<0.1	[29]
OTA	SPE	LC-FD	150	40	15 ^#^	0.266	[30]
OTA	SPE	LC-FD	100	-	5 ^#^	0.02	[31]
OTA	SPE	LC-FD	40	>60	10 ^#^	<0.01	[9]
OTA	SPE	LC-MSMS	2.5	6	0.25 ^#^	0.3	*

Note: SPE—Solid Phase Extraction; QuEChERS—Quick Easy, Cheap, Effective, Rugged, and Safe; MWE—Microwave Extraction; DLLME—Dispersive Liquid-Liquid Micro Extraction; DLLME-SFO—Dispersive Liquid-Liquid Micro Extraction Solidified Organic Droplet; LC-MSMS—Liquid Chromatography-Tandem mass spectroscopy; LC-FD—Liquid Chromatography Fluorescent detector; # Sample amount in grams; * Sample amount in mL.

## Data Availability

Data are unavailable due to privacy or ethical restrictions.

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
