# Peer review of "Rapid Identification and Analysis of Ochratoxin-A in Food and Agricultural Soil Samples Using a Novel Semi-Automated In-Syringe Based Fast Mycotoxin Extraction (FaMEx) Technique Coupled with UHPLC-MS/MS"

_molecules, 2023, doi:10.3390/molecules28031442_

Round 1

Reviewer 1 Report

Well written paper presenting a high-throughput extraction and analytical method to analyze mycotoxin (OTA) in coffee, tea, and soil samples.

Minor comments: Lines 27-32: too much details about the method in the abstract. It can be shortened.

Figure 2a: in the text, 5 solvents were tested but only 3 of them are shown. Please show all of them or explain why showing only 3.

Section 2.1. Optimization of the extraction process: Was this done with pure OTA in solvent or with samples? Please, explain your choice.  

Figure 3: Why are (a) and (b) scatter plots and not bar plots like (c) and (d)? Might be better to present everything in the same format

Section 2.3.1. Analytical performances of the developed method: Could we see the calibration curves? 

Table 1: are intraday and interday results in %? please explain how is it calculated.

Materials and Methods: Problem with the figure numbers: you already have 4 figures in the results and start again with Figure 3 and 4 in the M&M... please fix it.

Author Response

Well-written paper presenting a high-throughput extraction and analytical method to analyze mycotoxin (OTA) in coffee, tea, and soil samples.

Response: Authors would like to express our sincere thanks and gratitude to the expert reviewer for spending his/her valuable time in reviewing our manuscript and pointing out the critical scientific insights/comments to improve the overall quality of our manuscript. We have revised our manuscript deeply as per the expert reviewer’s comments/corrections. Revised portions are marked in the yellow background in the re-submitted R1 manuscript. Please find below our response point to point responses to reviewer' comments as below,

Minor comments: Lines 27-32: too much details about the method in the abstract. It can be shortened.

 Thanks for the suggestion to improve the abstract part, we have re-written the parts as follows:

Revised part as below:

In the extraction process, 0.25 g sample and extraction solvent were added to the first syringe barrel for extraction under vigorous vortex shaking for 2.5 min. Then, the extraction syringe was connected to a clean-up syringe barrel (pre-packed with clean-up sorbents/salt such as C18, activated carbon, and MgSO4) with a syringe filter for the clean-up process. Afterward, the whole set-up was placed in an automated programmable mechanical set-up for controlled elution at a constant flow rate. To enhance FaMEx technology performance, the various influencing parameters, including extraction solvents type and its volume, extraction time, clean-up sorbents, and elution flow rate) were optimized.

Figure 2a: in the text, 5 solvents were tested but only 3 of them are shown. Please show all of them or explain why showing only 3.

Thank you for suggesting the valuable comment in figure 2a, for testing the various extraction solvents we have studied the five solvents MeCN, ACTN, EA, MeOH, and IPA among the five solvents, only three of the solvents showed good extraction recovery. Presently we have included all 5 solvents as well as for the tea and soil as per the 3rd reviewer’s suggestions.

Figure 2: (a) Selection of extraction solvent, and (b) Selection of extraction time for the extraction of OTA from coffee samples. Extraction conditions: 250 mg sample, 2.0 mL solvent, 2.5 min of vortex extraction, and clean-up under the flow rate of 0.5ml/min using the 1000 mg MgSO4, 75 mg AC and 50 mg C18 with three replicate analyses.

Section 2.1. Optimization of the extraction process: Was this done with pure OTA in solvent or with samples? Please, explain your choice. 

Thank you for suggesting the valuable comment about the important part of the work, herein we optimized all the extraction parameters using coffee samples and we applied them to tea and soil samples; therefore, we performed the experiments by spiking the OTA in coffee samples for all the optimization process. And in the manuscript, we mentioned line number 95. Presently we have given the data for the various optimization steps as per the 3rd reviewer’s suggestions.

Figure 3: Why are (a) and (b) scatter plots and not bar plots like (c) and (d)? Might be better to present everything in the same format

Figure 1 and 2 is changed under the reviewer’s comments and updated in the manuscript.

Thank you for suggesting the valuable comment about figure 3. The scatter plots are changed to the bar plots and updated in the same in the manuscript.

Section 2.3.1. Analytical performances of the developed method: Could we see the calibration curves?

Thank you for suggesting the valuable review for the analytical method validation section. Here I have given the calibration curve for your view.

Figure: Calibration curve of the coffee/tea and soil samples.

Table 1: are intraday and interday results in %? please explain how is it calculated.

Thank you for suggesting the valuable comment in the interday and intraday results part. The terms are mistakenly written as interday analysis and intraday analysis, instead interday recovery and intraday recovery in percent. The equation for extraction recovery calculation and table 1 columns were corrected as interday recovery and intraday recovery in percent in the manuscripts as shown in the following.

Materials and Methods: Problem with the figure numbers: you already have 4 figures in the results and start again with Figures 3 and 4 in the M&M... please fix it.

Thank you for suggesting the valuable comment in the materials and methods section, we updated the figure number in the M&M section figures in the manuscript.

Modified changes in the manuscript:

Figure 5: Illustration of the design of the developed semi-automated instrument for the FaMEx clean-up process.

Figure 6: Graphical representation of FaMEx extraction procedure.

Response: Thank you very much for your valuable time to read this document. Your comments and feedbacks are highly beneficial to us in revising our work. As per your review comments, we have significantly revised our manuscript in detail, and proper referencing has been done. Hopefully, the revised manuscript is suitable and convinces the reviewer.

Reviewer 2 Report

In this manuscript, a semi-automated in-syringe based fast mycotoxin extraction technique coupled with UHPLC-MS/MS was developed for the rapid analysis of ochratoxin-A in food and agricultural soil samples. The parameters were optimized in detail. However, this manuscript can not be accepted in the present version. Some issues should be clarified.

1. In this manuscript, there are too many subjective descriptions of experimental results, such as the section of pH, extraction solvent volume, and so on. Figures should be added.

2. The authors have described the results in detail, but the explanations for these results are very limited.

3. Line 118-121, “poor chromatographic behavior” and “intense brown color after extraction” are subjective descriptions. At least, the chromatograms (though they are poor) and photos of extracts after extraction should be presented. If the number of figures is limited in the manuscript, the figures can be added in Supplementary.  

Author Response

In this manuscript, a semi-automated in-syringe-based fast mycotoxin extraction technique coupled with UHPLC-MS/MS was developed for the rapid analysis of ochratoxin-A in food and agricultural soil samples. The parameters were optimized in detail. However, this manuscript cannot be accepted in the present version. Some issues should be clarified.

Response: Authors would like to express our sincere thanks and gratitude to the expert reviewer for spending his/her valuable time in reviewing our manuscript and pointing out the critical scientific insights/comments to improve the overall quality of our manuscript. We have revised our manuscript deeply as per the expert reviewer’s comments/corrections. Revised portions are marked in the yellow background in the re-submitted R1 manuscript. Please find below our response point to point responses to reviewer' comments as below,

In this manuscript, there are too many subjective descriptions of experimental results, such as the section on pH, extraction solvent volume, and so on. Figures should be added.

Thank you for suggesting to include the figures in the manuscript, as per the suggestion pH optimization data was included in the supplementary data, which we presented here.

During the solvent addition of the extraction process, we found that 1 mL of solvent for the coffee and tea samples was not enough to get complete immersion, therefore we chose the higher volume of the solvent for the coffee and tea samples. Therefore, we didn’t get the complete data sets for the solvent volume optimization for three of the sample matrices.

Figure: Effect of various pH mediums for the extraction of Ochratoxin A from coffee, tea and soil samples. Extraction conditions: 250 mg coffee, 2.0 mL solvent (under different pH conditions), 2.5 min of vortex extraction, and clean-up under the flow rate of 0.5ml/min using 1000 mg MgSO4, 75 mg AC and 50 mg C18 with three replicative analysis.

The authors have described the results in detail, but the explanations for these results are very limited.

Thank you for the suggestions to include the result explanation. We have included the explanations in the two sections as follows.

2.1.1 Effect of extraction solvent

The extraction solvent is a significant factor that affects the extraction performance in FaMEx. Selecting a suitable extraction solvent is essential to avoid a negative influence on the extraction process. A proper solvent must possess high extraction capacity and good chromatographic behavior for OTA separation and detection [16]. Five organic solvents were screened, including acetonitrile (MeCN), ethyl acetate (EA), acetone (ACTN), methanol (MeOH), and isopropyl alcohol (IPA), and were tested for OTA extraction and analysis under the developed method. 2.0 mL of extraction solvent was taken and vortexed for 2.5 min at 3000 rpm, and the extraction solvent was passed through the FaMEx-SPE cartridge containing 50 mg C18, 75 mg AC, and 1000 mg anhydrous MgSO4 for the clean-up process. Acetonitrile (MeCN) shows maximum extraction recovery due to the least polar nature compared to the other selected solvents and which shows >90% with no matrix peaks interferences, as illustrated in Figure 2a. EA solvent extraction results indicated poor chromatographic behavior and improper peak shape (non-symmetrical) for OTA. MeOH, IPA, and ACTN extractions resulted in intense brown color after extraction due to the higher affinity towards the tannin, Millard products, chlorophylls, and flavonoids. This may be due to the hydrogen bonding interaction with the hydroxyl group in MeOH and IPA and the solvents result in the lowest extraction recovery for OTA. Thus, MeCN was eventually chosen as an extractant for subsequent analysis.

2.1.4 Effect of extraction (vortex) time

The extraction time is an essential factor in achieving the highest extraction of OTA from the sample matrix. Inversely, coffee, tea, and soil products are more affluent in Millard products, chlorophylls, lipids, proteins, flavonoids, caffeine, phenolic acids, and numerous degradation products; hence increase in the extraction time leads to co-extract the matrix components in the sample matrix tends to poor performances. Various extraction times (from 0.5 to 3 min) under a vortex agitation mixer were examined to achieve higher OTA extraction with low matrix interferences. Figure 2b indicates that extraction recovery of OTA was increased from 1 to 3 min and remained constant at 2.5 min. Therefore, 2.5 min of extraction (vortex agitation) time is ideal for extracting maximum OTA from the sample matrix and it can able to avoid the less acetonitrile soluble matrices such as lipids and unsaturated fat-ty acids [31]; hence, 2.5 min of extraction time was selected for the FaMEx-SPE clean-up optimization process.

Lines 118-121, “poor chromatographic behavior and “intense brown color after extraction” are subjective descriptions. At least, the chromatograms (though they are poor) and photos of extracts after extraction should be presented. If the number of figures is limited in the manuscript, the figures can be added in Supplementary.

Thank you for your valuable suggestions about the presentation of chromatograms in the manuscript. We presented the chromatogram and the final extraction solvents in the supplementary file as depicted below.

Figure S1, (a1-e1) Chromatogram of OTA extracted from coffee using acetonitrile, acetone, ethyl acetate, methanol, and isopropyl alcohol. (a2-e2) Color of coffee extracts from acetonitrile, acetone, ethyl acetate, methanol, and isopropyl alcohol.

Response: Thank you very much for your valuable time to read this document. Your comments and feedbacks are highly beneficial to us in revising our work. As per your review comments, we have significantly revised our manuscript in detail, and proper referencing has been done. Hopefully, the revised manuscript is suitable and convinces the reviewer.

Reviewer 3 Report

The publication is interesting, but its novelty is small. It should be published, but perhaps not in a journal with such a high IF as Molecules. I think the editor should suggest/help the authors move it to another MDPI journal with a lower IF (e.g., Separations).

Analogous adsorbents can be successfully used for SPE using 96-well-plates. The method would be faster compared to the method developed by the authors, which requires a special design with a syringe.

Line 117-118: the authors write 'fewer  matrix peaks interferences ' and refer to figure 2, which shows no peaks from the matrix.... please provide in a separate figure chprmatograms for extracts obtained for all solvents used. Then we will be able to assess the effects of asymmetry peaks, or peaks from the matrix, which the authors write about in the next sentence (lines 118-120)

Figure 2: please show analogous results for other matrices, not just coffee

Section 2.1.2: please tabulate the recoveries for each of the martens and for each of the solvents described herein

Section 2.1.3: was the same volume optimal for each matrix? and for soil and for food?

Section 2.1.4: please provide similar results for all other matrices mentioned here (coffee, tea, and soil products)

Chapter 2.2: I don't understand when a mixture of AC and C18 started to be used in the study. Is it a coincidence that the entire chapter 2.1 are not the results already obtained for a mixture of AC and C18 and magnesium sulfate? If not, for which adsorbent are the results presented in chapter 2.1?

What conditions were used for AG, GCB that for them a good purification was not obtained?

Lines 235-238: couldn't the authors have changed the chromatographic method to avoid/minimize coelution of compounds from the matrix with OTA? This is a common practice. Please show a sample chromatogram so that you can see how significant coelution there is.

Section 2.3.2: what is matrix-matched solvent in your case?

Figure 4: what kind of chromatograms do we have here? EIC?

Author Response

Authors would like to express our sincere thanks and gratitude to the expert reviewer for spending his/her valuable time in reviewing our manuscript and pointing out the critical scientific insights/comments to improve the overall quality of our manuscript. We have revised our manuscript deeply as per the expert reviewer’s comments/corrections. Revised portions are marked in the yellow background in the re-submitted R1 manuscript. Please find below our response point to point responses to reviewer' comments as below,

The publication is interesting, but its novelty is small. It should be published, but perhaps not in a journal with such a high IF as Molecules. I think the editor should suggest/help the authors move it to another MDPI journal with a lower IF (e.g., Separations). Analogous adsorbents can be successfully used for SPE using 96-well plates. The method would be faster compared to the method developed by the authors, which requires a special design with a syringe.

The presented manuscript is submitted to the Special Issue "Application of Liquid Chromatography in Food and Natural Products Extracts Analysis" in Molecules journal, based on the pre-approval and encouragement by the editorial team since the paper is fit to the special issue scope and its quality.

Moreover, researchers grabbed attention towards automating the QuEChERS methodology and particularly when it comes to complex matrices food samples analysis such as coffee and tea.  Therefore, our research group is proposing the present methodology which we use simple and low-cost plastic syringes for solvent extraction and for cleanup. And the syringe volumes can be varied based on the sample amount or extraction solvent volume. In addition, its easy to pack cleanup sorbents at various amounts in the syringe when compared to limited sorbent space in 96-well plate-based SPE.

Therefore, we believe that our work is simple, novel, low-cost, semi-automated, and efficient for food analysis.

Line 117-118: the authors write 'fewer matrix peaks interferences ' and refer to figure 2, which shows no peaks from the matrix.... please provide in separate figure chromatograms for extracts obtained for all solvents used. Then we will be able to assess the effects of asymmetry peaks, or peaks from the matrix, which the authors write about in the next sentence (lines 118-120)

Dear reviewer, we thank you for the suggestions for presenting the chromatographic figures for the different extraction solvents. In the manuscript, we reached the maximum number of figures therefore, we depicted chromatograms in the supplementary file, as shown below.

Figure S1, (a1-e1) Chromatogram of OTA extracted from coffee using acetonitrile, acetone, ethyl acetate, methanol, and isopropyl alcohol. (a2-e2) Color of coffee extracts from acetonitrile, acetone, ethyl acetate, methanol, and isopropyl alcohol.

Figure 2: Please show analogous results for other matrices, not just coffee

Thank you for suggesting the valuable comment for figure 2. As per your suggestions, we have depicted the other matrices results in the figure 2 in manuscript.

Figure 2: (a) Selection of extraction solvent, and (b) Selection of extraction time for the extraction of OTA from coffee, tea, and soil samples. Extraction conditions: 250 mg coffee, 2.0 mL solvent, 2.5 min of vortex extraction, and clean-up under the flow rate of 0.5ml/min using the 1000 mg MgSO4, 75 mg AC and 50 mg C18 with three replicate analyses.

Section 2.1.2: please tabulate the recoveries for each of the martens and for each of the solvents described herein

Thank you for your comment on section 2.1.2.  As per the suggestions from reviewers 2 & 3 and in consideration of the journal’s limit, we have submitted the recovery results in the bar graphs and have also presented them in the supplementary data. Please find the same for your perusal below.

Figure: Effect of various pH mediums for the extraction of Ochratoxin A from coffee, tea, and soil samples. Extraction conditions: 250 mg coffee, 2.0 mL solvent (under different pH conditions), 2.5 min of vortex extraction, and clean-up under the flow rate of 0.5ml/min using 1000 mg MgSO4, 75 mg AC and 50 mg C18 with three replicative analysis.

Section 2.1.3: was the same volume optimal for each matrix? and for soil and for food?

Thank you for your valuable suggestions in section 2.1.3. Yes, we selected the same solvent volume for the extraction of coffee, tea, and soil samples.

During the solvent volume test for the extraction process, we found that 1 mL of solvent for the coffee/ tea/soil samples was not enough to get complete extraction, therefore, based on the experimental optimization data, we have chosen the higher volume of the solvent for all the samples.

Section 2.1.4: Please provide similar results for all other matrices mentioned here (coffee, tea, and soil products)

Thank you for your valuable suggestions in section 2.1.4. As per your suggestions, we have depicted the other matrices’ results in figure 2 in the manuscript.

Figure 2: (a) Selection of extraction solvent, and (b) Selection of extraction time for the extraction of OTA from coffee, tea, and soil samples. Extraction conditions: 250 mg coffee, 2.0 mL solvent, 2.5 min of vortex extraction, and clean-up under the flow rate of 0.5ml/min using the 1000 mg MgSO4, 75 mg AC and 50 mg C18 with three replicate analyses.

Chapter 2.2: I don't understand when a mixture of AC and C18 started to be used in the study. Is it a coincidence that the entire chapter 2.1 are not the results already obtained for a mixture of AC and C18 and magnesium sulfate? If not, for which adsorbent are the results presented in chapter 2.1?

Thank you for your valuable suggestion regarding the overall optimization of the extraction step. Based on the literature survey, we have selected the sorbents for the extraction process, and we have pre-optimized the extraction steps for AC and C18. We have also tried other commercial sorbents for the extraction steps in the clean-up optimization stage, which also exhibited good results under the same sorbent conditions as chosen first.

What conditions were used for AG, and GCB that for them good purification was not obtained?

Thank you for your valuable comments about the selection of various sorbents, for the overall extraction technique to determine the OTA in coffee, tea, and soil samples, we performed one variant at a time approach. For the selection of various sorbents, we optimized the various extraction process and as mentioned in the manuscript, later when it comes to sorbent optimization, the extractant was cleaned up separately using different amounts of AG and GCB, both of the sorbents were not yielded the inconsistent results. The extraction conditions used for the analysis are 250 mg coffee, 2.0 mL solvent, 2.5 min vortex extraction, and clean-up under the flow rate of 0.5mL/min using the 1000 mg MgSO4, 50 mg of AG or 50 mg of GCB with three replications.

Lines 235-238: couldn't the authors have changed the chromatographic method to avoid/minimize coelution of compounds from the matrix with OTA? This is a common practice. Please show a sample chromatogram so that you can see how significant coelution there is.

Thank you for your valuable comments on the matrix effect in the present work. As mentioned in the manuscript, three of the concentrations were spiked in the matrix-matched solvent. The calculated matrix effects were -13.77 ± 7.31 for coffee and -9.7 ± 6.23 for soil samples. Since we used tandem mass spectrometer as the detector, we didn’t see any other peaks for the OTA signal suppression, instead, the peak area was reduced because of the matrix interference and the representative total ion chromatograms of the blank, and 1 ng/g OTA extracted from coffee, tea, and soil matrix were depicted in figure 4 as well as the chromatogram of various extracted solvent is given in the figure S1.

Figure 4. Total ion chromatogram of blank solvent, and 1 ng/g OTA extracted from coffee, tea, and soil matrix.

Section 2.3.2: what is matrix-matched solvent in your case?

Thank you for your valuable comments in section 2.3.2; here we obtained the matrix-matched solution under the FaMEx protocol for blank coffee/tea and soil samples to collect the 10 mL of pooled solvents separately for different matrix and used it as a matrix-matched solvent for the analysis.

Figure 4: what kind of chromatograms do we have here? EIC?

Thank you for your valuable questions, here we presented the total ion chromatogram of blank solvent, and 1 ng/g OTA extracted from coffee, tea, and soil matrix. In manuscript we fixed the issues in the figure captions as well as in the in the section 2.3.3.

Thank you very much for your valuable time in reading this document. Your comments and feedbacks are highly beneficial to us in revising our work. As per your review comments, we have significantly revised our manuscript in detail, and proper referencing has been done. Hopefully, the revised manuscript is suitable and convinces the reviewer.

Round 2

Reviewer 3 Report

If the editor has decided that he does not want to reject this publication, and wants it to be accepted for a special issue, I have no choice but to accept his decision.